
# 1  Bushfire effects on soil properties and post-fire slope stability: the
# 2  case of the 2015 Wye River-Jamieson Track bushfire

Yuanying Li[1], Akihiko Wakai[1], Susanga Costa[2]
[1]Graduate School of Science and Technology, Gunma University, Kiryu, 3768515, Japan
[2]School of Engineering, Deakin University, Waurn Ponds, 3216, Australia
*Correspondence to*: Yuanying Li (t222c001@gunma-u.ac.jp)
**Abstract.** Bushfire is a destructive natural disaster that leads to vegetation loss and increased soil infiltration. Over a long
post-fire period, root death and reduced reinforcement decrease soil shear strength. During rainfall, shallow landslides in
burned areas become more frequent and widespread. This study focused on Wye River and Separation Creek in Australia,
affected by the 2015 Wye River-Jamieson Track bushfire. Ten months after the bushfire, multiple slope failures, including
the Paddy's Path landslide, occurred during heavy rains from 12 to 14 September 2016, disrupting the Great Ocean Road
connecting towns. This study aims to assess changes in slope stability during rainfall before and after the bushfire.
Controlled laboratory burning tests simulated bushfire effects on soil, resulting in changed soil properties after the fire:
increased permeability due to soil particle coarsening and reduced soil shear strength, especially cohesion. Considering the
changes in soil properties before and after the fire, a simplified hydrological numerical model for infiltration calculation was
employed to analyze time-dependent changes in groundwater level depth, surface water depth, and safety factor during
rainfall. Comparing pre- and post-fire results indicated higher susceptibility to shallow slope failures in burned areas, with
rapid rises in groundwater level and surface water acting as triggers. These findings enhance the understanding of landslide
triggering mechanisms in post-fire slopes and provide insights for mapping landslide susceptibility in bushfire-prone regions.

## 20  1 Introduction

Bushfires, also known as wildfires, are prevalent natural disasters that burn approximately 350 million ha globally each year
and are intensified by global warming and drought (Giglio et al., 2013). Particularly affected regions include the
Mediterranean, the Amazon, the western United States, and south-eastern Australia (Bowman et al., 2020). In Australia,
bushfires cost about 8.5 billion dollars annually, roughly 1.15 % of GDP, with severe fires in recent decades in Sydney,
Canberra, and Melbourne's exurban margins, causing significant losses of life, property, and forest resources (Ashe et al.,
2009). Bushfires not only cause immediate losses but also lead to lasting hydrological and geomorphological changes,
affecting soil physicochemical properties for months to years (Shakesby and Doerr, 2006). Post-fire vegetation losses
enhance infiltration, and reduce evapotranspiration, which elevate pore water pressure during rainfall and lower the threshold
for landslides (Staley et al., 2017). Lainas et al. (2016) reported that burned slopes require 20 %-30 % less rainfall to trigger




landslides compared to unburned ones. In the early 21st century, frequent rainfall-induced landslides and debris flows in
post-fire eucalypt forests in south-eastern Australia caused socioeconomic losses exceeding 60 million dollars (Freund et al.,
2017). Given the severe impact of these events, proactive assessment and mitigation strategies are crucial. Despite the
extensive damage and the clear need for preventive measures, studies on post-fire landslides remain limited (Abdollahi et al.,

34  2023).


Bordoloi and Ng (2020) indicated that despite the significant increase in the scale and frequency of bushfires, few studies
have addressed changes in soil mechanical properties and their impacts on post-fire slope stability. Identifying changes in
soil properties caused by bushfires remains challenging (Certini, 2005). Although it is acknowledged that changes in soil
physical and hydraulic properties due to fires are related to slope stability, the exact nature and extent of these changes
depend on factors such as fire severity, ecoregion, and time since the fire (Akin et al., 2023). In situ tests after bushfires have
proven instructive for collecting post-burn soil data, yet field investigations present various difficulties (Moody et al., 2013).
Unstable terrain and road closures immediately following bushfires hinder data collection. Fire sites may contain hazards
such as embers and unstable trees, posing risks to researchers (Brucker et al., 2022). Since soil properties are influenced by
factors such as topography, soil type, vegetation regimes, and climatic conditions, post-fire responses vary regionally,
leading to geographical variability in field survey results (Agbeshie et al., 2022). Additionally, the frequent absence of pre-
fire soil data restricts comparative analyses of slope stability before and after fires, making it challenging to accurately assess
the impact of changes in soil properties on post-fire stability.

Laboratory burning tests provide a highly controllable method for evaluating and quantifying changes in soil properties due
to bushfire impacts (Wieting et al., 2017). Previous laboratory studies have simulated bushfire effects on soils, examining
changes in properties such as particle size, saturated hydraulic conductivity, and infiltration capacity (Badía and Martí,
2003). Controlled settings in these tests overcome logistical challenges in field research, ensuring a safe working
environment and enabling precise control over experimental conditions and repeatability, which are difficult to achieve in
natural settings (Babrauskas and Grayson, 1990; Brucker et al., 2022). This approach allows for effective simulation of
diverse environmental conditions, such as various soil types, fire intensities, and specific regional conditions. It can study
how these environmental factors individually and interactively affect soil properties, enhancing understanding of variability
and causality (Fontúrbel et al., 2012). Despite the significant spatial and temporal variability in data on site, laboratory
burning tests can more accurately analyze the impact of fires on soil by controlling variables, thus minimizing the
interference of geographical variations in research outcomes. Such laboratory simulations replicate natural bushfire
mechanisms, offering an alternative analytical technique (Pereira et al., 2019). Therefore, laboratory burning tests are
recognized for their significant advantages in safety, controllability, reproducibility, and validity when determining changes
in soil properties before and after bushfires compared to investigations conducted on site.





Assessing post-fire landslides requires studying changes in the force balance of slope soils. Landslides occur when shear
stress exceeds the shear strength of slope materials (Sidle and Ochiai, 2006). Previous research on post-fire slope stability
has primarily focused on erosion or debris flow initiation using historical data, empirical studies, or statistical models. Staley
et al. (2017) defined new logistic regression equations for predicting debris-flow likelihood from empirical databases and
refined geospatial analysis. Gartner et al. (2008) developed an empirical model to estimate debris-flow volumes across
diverse regions and geological contexts. Compared to these statistical approaches, limited studies have directly used methods
based on the formation mechanisms of post-fire slope failures. Few studies on shallow landslides after bushfires have
identified the primary triggering mechanism as the loss of structural support from roots (Akin et al., 2023). This loss results
in decreased soil shear strength and increased pore water pressure following wetting events due to rising groundwater levels,
which are highly correlated to landslide occurrence after bushfires. Furthermore, landslides can occur immediately after the
first rainy season following a fire or months to years later (Cannon and Gartner, 2005; Meyer et al., 2001; May and
Gresswell, 2003). To establish changes in slope stability before and after fires, it is necessary to differentiate the effects of
the fire from triggering meteorological conditions, such as rainfall.

Understanding post-fire slope stability during rainfall is crucial for mitigating hazards. The Wye River-Jamieson Track
Bushfire, which began on 19 December 2015, caused severe eucalyptus vegetation loss in the townships of Wye River and
Separation Creek along the Great Ocean Road in the Otway Ranges, Australia. Ten months after the bushfire, heavy rain
caused landslides that temporarily closed the road, impeding recovery efforts (Colls and Miner, 2021). This study is based on
rainfall data and slope failures that occurred during this period. The loss of vegetation, reduced soil shear strength, and
increased pore water pressure are considered key factors for the mechanism of landslide occurrences in burned slopes. This
study aims to assess changes in slope stability during rainfall before and after the fire. For this purpose, soil samples were
taken from unburned locations near the burned area, and controlled laboratory burning tests were conducted to simulate
bushfire effects on soils. Measurements of soil texture, infiltration capacity, and shear strength were performed before and
after burning, with pairwise comparisons. Using a simplified hydrological approach-based numerical method (Wakai et al.,
2019; Ozaki et al., 2021; Nguyen et al., 2022), this study assessed slope stability during rainfall by comparing groundwater
level depth, surface water depth, and safety factor before and after the bushfire. The findings contribute to understanding
landslide-triggering mechanisms in post-fire slopes and provide insights for land management and disaster prevention in fire-
prone regions.
**2 Scope of the study**
**2.1 The 2015 Wye River-Jamieson Track bushfire and subsequent observed slope failures**
The 2015 Wye River-Jamieson Track bushfire, ignited by lightning strikes in the Great Otway National Park, 130 km
southwest of Melbourne, Australia, erupted on 19 December 2015 (hereafter referred to as the 2015 bushfire). Severe
weather caused the fire to breach containment lines on 24 December, impacting Wye River and Separation Creek. The 2015
bushfire destroyed 98 houses in Wye River and 18 houses in Separation Creek, prompting emergency alerts. It burned for 34
days until containment on 21 January 2016, scorching over 2500 ha (Fig. 1a). Economic losses were estimated at over 60
million dollars. The affected regions, covered by flammable eucalyptus forests, experienced record heat in December 2015
and below-average rainfall, which increased fuel levels (State Government of Victoria, 2016). Wye River and Separation
Creek are located on steep, southeastern slopes surrounded by rugged terrain, posing challenges for firefighting and
evacuation. As two adjacent towns, Wye River and Separation Creek are connected by the Great Ocean Road. They lie in the
Otway Ranges, one of Australia's most landslide-prone regions, with significant landslide events typically linked to high
annual rainfall (Dahlhaus and Miner, 2002). The Paddy's Path landslide was first observed in Wye River and Separation
Creek following a daily rainfall of approximately 60 mm on 14 September, the highest recorded in 2016. The Paddy's Path
landslide covered an area approximately 30 m wide by 40 m long (Colls and Miner, 2021), leading to the closure of the
Great Ocean Road for three weeks and months of traffic restrictions. This study aims to assess slope stability after the 2015
bushfire, focusing on slope failures following rain events around 14 September 2016. The study location (latitude: 38°37'31"
S, longitude: 143°54'24" E) was determined by comparing aerial photographs from Google Earth before and after the 2015
bushfire and subsequent rain events (Fig. 1a). This study area includes major residential areas and roads like Paddy's Path
and the Great Ocean Road, where slope failures occurred (Fig. 1b).

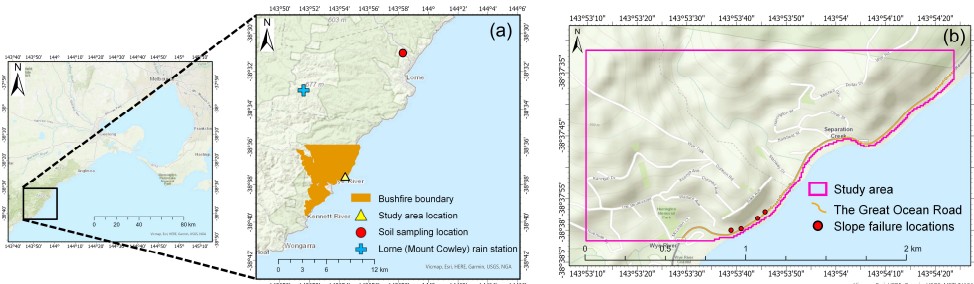


**Figure 1: (a) Boundary affected by the 2015 bushfire, and locations of the study area and soil sampling (adapted from Colls and**
**Miner, 2021). (b) Study area and slope failure locations. (from ESRI)**
**2.2 Soil sample collection**
Bushfires lead to changes in ground material properties, potentially reducing slope stability. However, the recovery time of
soil properties after a bushfire remains unclear (Akin et al., 2023). To compare slope stability before and after the 2015
bushfire, soil sampling was conducted in Lorne, an area unaffected by the 2015 bushfire. This sampling location, 16 km
from the study area, has similar climatic and rainfall conditions, and the same types of rock and soil (Fig. 2). The sampling
location in Lorne shows no signs of vegetation or soil burn (Fig. 3). Both disturbed and undisturbed soil samples were
collected: undisturbed samples using cylindrical and block sampling techniques, and disturbed samples from depths


exceeding 10 cm below the ground surface. Soil burning tests in laboratory were employed to simulate bushfire impact and
understand changes in post-fire soil properties.

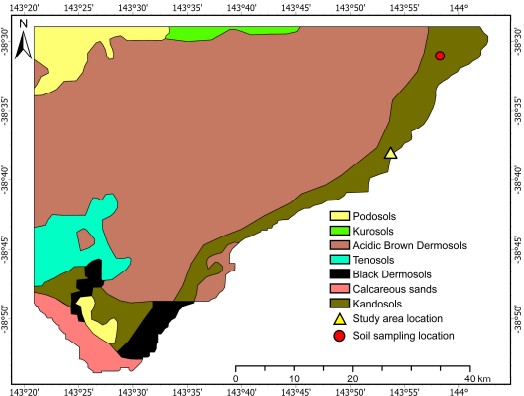


**Figure 2: Soil map of the study area and sampling location (using ESRI adapted from Agriculture Victoria, 2014).**

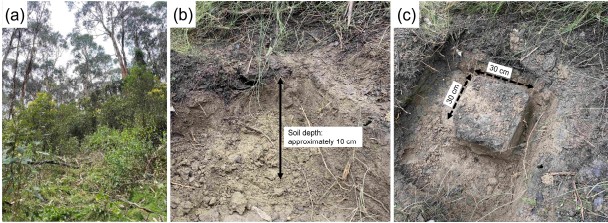


**Figure 3: (a) Unburned slope for soil sampling. (b) Soils on site, with disturbed samples taken from soil depths exceeding 10 cm. (c)**
**In situ block soil sample (L: 30 cm; W: 30 cm; H: 30 cm).**
**3 Laboratory burning test for bushfire simulation**
**3.1 Laboratory burning test conditions**
Laboratory simulations, often using muffle furnaces, are commonly employed to study bushfire effects on soil properties by
imitating the direct impact of temperature on soil (Pereira et al., 2019). This technique has proven effective in simulating
bushfire effects (Galang et al., 2010). Peak temperature and duration of burning are the main factors influencing soil
changes, necessitating careful control of test conditions (Pereira et al., 2019). However, obtaining retrospective soil
temperature data is challenging. Soil temperatures during bushfires typically range from 25 to 900 °C, and high-severity
burns reach up to 800 °C, consuming all litter and leaving bare earth (Soto et al., 1991; Mataix-Solera et al., 2011). Peak
temperatures as high as 1100 °C have been recorded worldwide (Goudie et al., 1992). Thus, the peak temperature of
bushfires is generally thought to be above 800 °C. According to the burn severity map of the 2015 bushfire (Noske et al.,


2022), most of the study area, particularly where slope failures were observed, experienced high-severity burning, resulting
in the destruction of all understory vegetation and scorching or burning of the canopy. Consequently, the maximum
temperature for the burning test was set at 800 °C. To prevent soil splashing, the samples were dried in an oven at 100 °C
overnight before placement in the furnace. The dried soil samples reached 800 °C in 1 h. Following the standard procedure
for laboratory burning tests (Giovannini, 1994), the samples were maintained at 800 °C for 30 min. After cooling to the
laboratory temperature around 20 °C, the samples were examined for changes in soil properties. Following these procedures,
laboratory burning tests were conducted on both disturbed and undisturbed soil samples to simulate post-fire soil conditions
(Fig. 4). A series of soil parameters were measured before and after burning, and the effects of the bushfire on soil properties
are described in the following sections and summarized in Table 1.

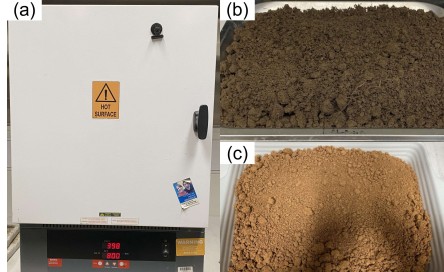


**Figure 4: (a) The furnace. (b) Unburned soil. (c) Burned soil.**
**Table 1: Summary of test results.**

| Test results | Soil conditions | |
|---|---|---|
| | Unburned | Burned |
| Moisture content, $w$ (%) | 18.93 | / |
| Wet density, $\rho_t$ (g cm$^{-3}$) | 1.71 | / |
| Dry density, $\rho_d$ (g cm$^{-3}$) | 1.44 | 1.42 |
| Soil particle density, $\rho_s$ (g cm$^{-3}$) | 2.67 | 2.66 |
| Void ratio, $e$ | 0.85 | 1.20 |
| Liquid limit, $w_L$ (%) | 23.19 | 0 |
| Plastic limit, $w_P$ (%) | 18.65 | 0 |
| Plasticity index, $I_p$ (%) | 4.55 | 0 |
| Hydraulic conductivity, $K$ (m s$^{-1}$) | 2.16E-07 | 9.15E-06 |
| Internal friction angle, $\varphi$ (deg.) | 39.33 | 40.24 |
| Cohesion, $c$ (kN m$^{-2}$) | 6.41 | 0.45 |




**3.2 Soil characteristics before and after burning**
**3.2.1 Changes in soil texture**
Soil texture represents the particle size distribution in the soil. Disturbed soil samples were burned following the procedures
in Sect. 3.1 to obtain burned soil. Grain size accumulation curves were generated using particle size distribution tests to
illustrate changes in soil texture before and after burning (Fig. 5). According to the Particle Size Classification in the
Australian Standard (Standards Australia, 2009), the soil was classified as fine sand (particle size limits: 200 μm to 60 μm)
before burning and as coarse sand (particle size limits: 2 mm to 600 μm) after burning. The curves indicate a decrease in fine
particle content and an increase in coarse particle content after burning, suggesting soil texture coarsening due to particle
bonding from heat. The primary reason for this phenomenon is that clay particles have a lower temperature threshold than
sand particles, making them more susceptible to change during fires (Neary et al. 2005).

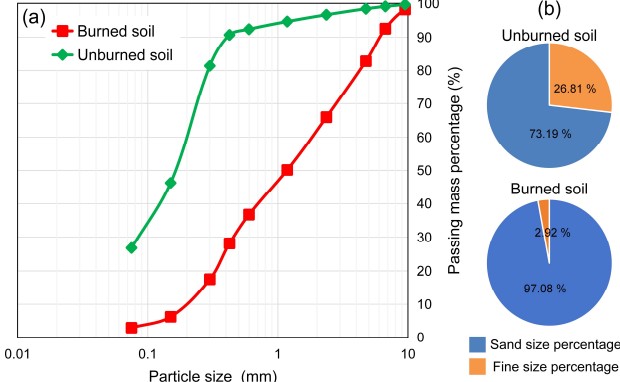


**Figure 5: (a) Grain size accumulation curve. (b) Particle size percentage for unburned and burned soils.**
**3.2.2 Changes in physical soil properties**
Physical properties of soils before and after burning were obtained using disturbed and undisturbed soil samples (Figs. 4 and
6), including moisture content, density, void ratio, and soil particle density (Table 1). The void ratio was calculated from dry
density and soil particle density. The results for moisture content, dry density, and wet density before burning were obtained
from undisturbed soil samples in cylindrical container A (Diameter: 62 mm; Height: 20 mm) shown in Fig. 6a. The value of
dry density after burning was determined from undisturbed soil samples in cylindrical container B (Diameter: 82 mm;
Height: 42 mm) shown in Fig. 6b-c. Comparison of the results confirms a slight decrease in dry density and soil particle
density after burning. This is consistent with the change in soil texture due to increased coarse particles. Therefore, the
increase in particle size from burning resulted in a higher void ratio, and reduced plasticity and elasticity of burned soils
were demonstrated based on the results of soil consistency limits. Giovannini et al. (1988) reported that high temperatures
promote the dispersion of soil aggregates, leading to a loss of stability.





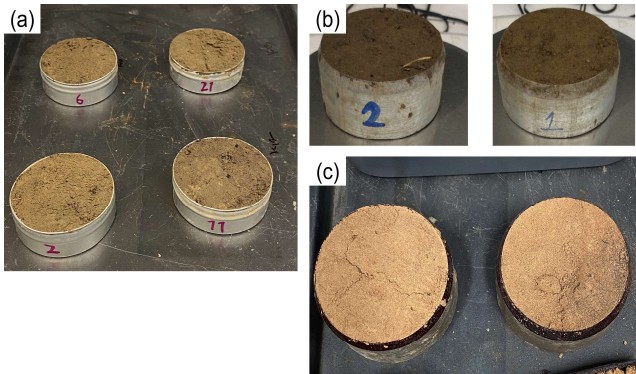


**Figure 6: (a) Cylinder sample A. (b) Unburned cylinder sample B. (c) Burned cylinder sample B.**

**3.2.3 Changes in soil hydraulic conductivity-$K$**
Soil hydraulic conductivities were measured on undisturbed block soil samples (Fig. 7; L: 20 cm; W: 18 cm; H: 20 cm)
before and after burning using the Mini Disk Infiltrometer (manufactured by METER Group, Inc., USA). The results shown
in Fig. 8 indicate a significant increase in the permeability of burned soils, due to increased soil porosity caused by larger
particle sizes.

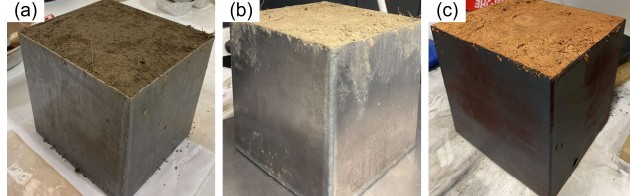


**Figure 7: (a) Wet block soil sample. (b) Dry block soil sample. (c) Burned block soil sample.**

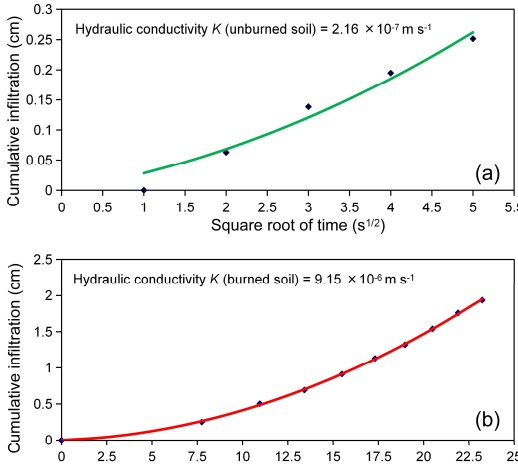


**Figure 8: Infiltrometer test results of hydraulic conductivity (a) before and (b) after burning.**

**3.2.4 Changes in soil mechanical properties-$c$, $\varphi$**
The results of shear strength parameters, $c$ and $\varphi$, were obtained from direct shear tests conducted using disturbed soil
samples before and after burning (Fig. 9). Comparison of the results reveals that while the internal friction angle ($\varphi$) did not
change considerably after burning, the cohesion ($c$) was significantly reduced. Sect. 3.2.1 demonstrates that burning enlarged
soil particles, changing the soil type from fine sand to coarse sand. This disruption of the original soil structure reduced
cohesion, resulting in decreased shear strength.

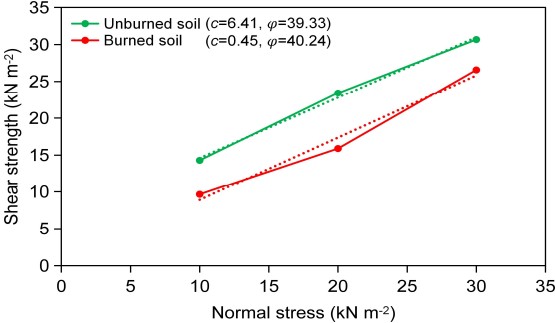


**Figure 9: Results of direct shear test**

## 4 Numerical method

### 4.1 Profile of the model

This study utilized a simplified hydrological method to assess slope stability (Wakai et al., 2019; Ozaki et al., 2021; Nguyen et al., 2022). The method assumes that the slope comprises a shallow impermeable layer (bedrock) overlain by permeable and homogeneous surface soil, allowing it to simulate various rainfall infiltration mechanisms due to differing permeability. Based on particle size distribution test results, the pre-bushfire slope was simulated using the fine sand model, which is characterized by relative low permeability. The post-bushfire slope, with increased permeability, was modelled using the medium-coarse sand model. The method involves four main components: calculating rainfall infiltration, predicting groundwater level fluctuations, modelling surface water, and performing a simple stability calculation using the semi-infinite slope assumption. Subsequent sections provide brief descriptions of these components.

### 4.2 Calculating rainfall infiltration

The first step involves calculating rainfall infiltration, separating rainfall into infiltration from the ground surface and surface runoff at each time. Green and Ampt (1911) proposed a simplified equation for one-dimensional vertical rainfall infiltration into a uniform soil profile under surface ponding conditions. This study used the modified Green-Ampt model by Mein and Larson (1973) and Chu (1978) for infiltration analysis, conforming to the homogeneous-slope assumption and the slope angle influence described by Wakai et al. (2019). The modified equation can accommodate changes in rainfall over a short time interval ($\Delta t$), as follows:

$$f(t + \Delta t) = K_s[1 + \Delta h \Delta \theta / F(t + \Delta t)] , \qquad (1)$$

$$F(t + \Delta t) = F(t) + K_s \Delta t + (\Delta h \Delta \theta) ln\left(\frac{F(t+\Delta )+\Delta h \Delta \theta}{F(t)+\Delta h \Delta \theta}\right) , \qquad (2)$$

$$F_p = K_s(\Delta h \Delta \theta)/(R(t) - K_s) , \qquad (3)$$

Where $f(t + \Delta t)$ is infiltration capacity at time $t + \Delta t$; $F(t)$ and $F(t + \Delta t)$ are cumulative infiltration at time $t$ and $t + \Delta t$, respectively; $\Delta \theta$ is change in volumetric water content; $K_s$ is saturated hydraulic conductivity; $R(t)$ is rainfall intensity; $F_p$ is cumulative infiltration at the start of ponding; and $\Delta h = (h + \psi_f)$ is the driving matric pressure head, where $h$ is the surface water depth and $\psi_f$ is the matric suction at the wetting front.

The infiltration capacity ($f$) determines the amount of infiltration water ($I$) at each step. If the total amount of precipitation ($R$) and surface water ($SW$) is less than the infiltration capacity ($f$), all the water will be absorbed into the soil. If the total amount exceeds the infiltration capacity ($f$), the infiltration water ($I$) equals the infiltration capacity ($f$).


$$I = \begin{cases} SW + R, \ if \ (SW + R) \leq f \\ f, \ if \ (SW + R) > f \end{cases}, \tag{4}$$

**4.3 Predicting fluctuations in groundwater level**

The second step involves modelling the spatial distribution of groundwater, which includes vertical infiltration from the ground surface and lateral infiltration of groundwater in the saturated zone of the slope. The predictive model in this study assumes that fluctuations in shallow groundwater levels of natural slopes are calculated based on a single geological composition. Materials are categorized as medium-coarse sand and fine sand due to their significantly different infiltration characteristics. This section briefly explains the proposed models for medium-coarse sand and fine sand. The medium-coarse sand model, developed by Nguyen et al. (2022), predicts groundwater rise on natural slopes at a relatively shallow depth. To determine the analytical parameters for this method, parametric studies were conducted using the VGFlow model (Cai and Ugai, 2004), controlling slope conditions such as permeability, slope angle, and initial moisture. The prediction for groundwater level rise is simply assumed to be divided into two periods. In the first period, the degree of saturation increases at a nearly constant rate without a groundwater level rise until the unsaturated area receives more rainwater than its capacity. In the second period, the groundwater level rises at a nearly constant rate. The modelling of the vertical infiltration process focuses on these two periods. The elapsed time ($t_1$) before the groundwater level begins to rise is determined as follows.

$$t_1 = \frac{h}{I}\left(\theta_{cp} - n\frac{S_{r0}}{100}\right), \tag{5}$$

Where $I$ is infiltration water; $n$ is soil porosity; $h$ is the initial thickness of the unsaturated layer; $S_{r0}$ is the initial degree of saturation at the start of rainfall; and $\theta_{cp}$ is the limit of the mean volumetric water content in the unsaturated layer before the groundwater table starts to rise.

The rise velocity of the groundwater level ($v_{wl}$) after it starts rising steadily is defined by the theoretically required amount of water to saturate the pores in the unsaturated layer, with an adjustment parameter $\alpha_v$ to match results (Wakai et al., 2019). Assuming the critical degree of saturation corresponding to the start of groundwater level rise is $S_r^*$, the elapsed time ($t_2$) before the groundwater level reaches the ground surface is defined by the following equation.

$$t_2 = \frac{h}{v_{wl}} = \frac{n\left(1-\frac{S_r^*}{100}\right)h}{\alpha_v I}, \tag{6}$$

Equations (5) and (6) are applied to calculate changes in groundwater level due to vertical infiltration during rainfall events. A planar flow analysis model for shallow groundwater is essential when lateral groundwater inflow/outflow affects slope stability. The governing equation for the seepage field, considering only lateral seepage flow in the unconfined aquifer, is defined by Eq. (7) (Japanese Association of Groundwater Hydrology, 2010).





$\quad \frac{\partial}{\partial x}\left(T_{xx}\frac{\partial \Phi}{\partial x}\right) + \frac{\partial}{\partial y}\left(T_{yy}\frac{\partial \Phi}{\partial y}\right) = n_e \frac{\partial \Phi}{\partial t},$ (7)
Where $T_{xx}$ and $T_{yy}$ are transmissivities in the x- and y-axis directions in planar coordinate systems, respectively; $n_e$ is
effective porosity; and $\Phi$ is the total head of groundwater. The explicit method proposed by Kinzelbach (1986) can be
employed to differentiate the equation.

In the medium-coarse sand model, the saturation process moves upward from the bedrock to the surface. Different from the
medium-coarse sand model, which predicts groundwater level rise during rainfall, the fine sand model proposed by Ozaki et
al. (2021) focuses on simulating the downward movement of the high-saturation zone (wetting front) from the ground
surface. Ozaki et al. (2021) theoretically determined the downward velocity ($v_{bs}$) of the wetting front based on the water
required to saturate the pores in the unsaturated layer. However, in practical applications, slightly different values can be
obtained depending on conditions. The fine sand model uses Eq. (8) to calculate the downward velocity, multiplied by the
correction factor ($\beta_v$) for generalization.
$\quad v_{bs} = \frac{\beta_v I}{n\left(1-\frac{S_{r0}}{100}\right)},$ (8)
Where $n$ is soil porosity; $S_{r0}$ is the initial degree of saturation at the start of rainfall; and $I$ is a constant rainfall intensity.

The elapsed time ($t_3$) from the start of rainfall until all the unsaturated layers (vertical thickness: $h$) below the groundwater
level reach saturation and the infiltration front reaches the initial groundwater level can be calculated using Eq. (9). If there is
no groundwater level, the infiltration front reaches the undrained edge at the bottom of the analysis area. The fine sand
model focuses solely on the vertical rainwater infiltration component and does not account for the horizontal component.
$\quad t_3 = \frac{h}{v_{bs}} = \frac{n\left(1-\frac{S_{r0}}{100}\right)h}{\beta_v I},$ (9)

### 268   4.4 Modelling surface water

The surface water model uses the shallow-water equations, derived from the equations of conservation of mass and linear
momentum (Navier–Stokes equations) (Di Giammarco et al., 1996).
Law of conservation of mass:
$\quad \frac{\partial h}{\partial t} + \frac{\partial (hu)}{\partial x} + \frac{\partial (hv)}{\partial y} = i$ (10)
The Navier–Stokes equations:
$\quad \frac{\partial (uh)}{\partial t} + \frac{\partial (hu^2)}{\partial x} + \frac{\partial (huv)}{\partial y} + gh(S_{fx} + \frac{\partial H}{\partial x}) = 0 \,,$ (11)


$\frac{\partial(vh)}{\partial t} + \frac{\partial(hu)}{\partial x} + \frac{\partial(hv^2)}{\partial y} + gh\left(S_{fy} + \frac{\partial H}{\partial y}\right) = 0$,     (12)
Where $H(x,y,t)$ is water surface elevation above a horizontal datum; $h(x,y,t)$ is local water depth; $t$ is time; $x$ and $y$ are
horizontal coordinates; $u(x,y,t)$ and $v(x,y,t)$ are flow velocities in x and y directions; $i(x,y,t)$ is net input rainfall;
$S_{fx}(x,y,t)$ and $S_{fy}(x,y,t)$ are friction slopes in the x and y directions; and $g$ is gravitational acceleration.

To simplify the complex equations, the insignificant influence components are removed (Di Giammarco et al., 1996; Zhu et
al., 2020). They are replaced by the diffusion wave approximation equations, written as Eqs. (13) and (14).
$S_{fx} + \frac{\partial H}{\partial x} = 0$,     (13)
$S_{fy} + \frac{\partial H}{\partial y} = 0$,     (14)
These equations are solved using the finite-difference technique in the MIKE SHE model (DHI, 2007). The change in water
depth from time step $t$ to $t + \Delta t$ can be calculated as:
$\Delta h = h(t + \Delta t) - h(t) = i \cdot \Delta t + \frac{\sum Q \Delta t}{\Delta x^2}$,     (15)
Where $\sum Q$ represents the total flow volume from four directions at the calculated node. The discharge from an upstream
node to a downstream neighbor is approximately calculated in Eq. (16):
$Q = \frac{K\Delta x}{\Delta x^{\frac{1}{2}}}(H_U - H_D)^{1/2} h_u^{5/3}$,     (16)
Where $(H_D, h_D)$ and $(H_U, h_U)$ are the sets of the local water level and water depth of the downstream and upstream nodes,
respectively; $h_u$ is the free water depth of the upstream node that can flow into the downstream neighbor; and $K$ is the
Strickler coefficient, calculated as the inverse of the Manning coefficient, reflecting the roughness of the ground surface in
the calculation.
**4.5 Performing simple stability calculation using the semi-infinite slope assumption**
When soil thickness is considerably smaller than slope length, an infinite plane slope can suitably approximate a hillslope
(Montgomery and Dietrich, 1994). This study uses the Mohr-Coulomb failure law to assess slope stability, considering shear
strength (soil resistance to shearing) and shear stress (downslope component of soil weight) along the potential failure plane.
The safety factor ($F_s$) is the ratio of shear strength to shear stress. The sliding surface is assumed planar, infinitely extended,
and coinciding with the interface between the soil cover layer and the impermeable layer. Slope stability is considered
unstable when $F_s$ is less than 1.0. Differences in the definitions of $F_s$ between medium-coarse sand (Eq. 17) and fine sand
(Eq. 18) arise from the consideration of water pressure on the sliding surface, based on the semi-infinite slope assumption.



For the medium-coarse sand model:
$$F_s = \frac{\tau_f}{\tau} = \frac{c' + [\gamma_t \cdot h_1 + (\gamma_{sat} - \gamma_w)h_2]\cos^2\theta \cdot \tan\varphi'}{(\gamma_t \cdot h_1 + \gamma_{sat} \cdot h_2)\sin\theta \cdot \cos\theta},$$ (17)
For the fine sand model:
$$F_s = \frac{\tau_f}{\tau} = \frac{c' + \gamma_{sat} \cdot H \cdot \cos^2\theta \cdot \tan\varphi'}{\gamma_{sat} \cdot H \cdot \sin\theta \cdot \cos\theta}$$ (18)
Where $\tau$ is the shear stress due to the sliding direction component of gravity of soil; $\tau_f$ is the shear strength of soil, which is
the maximum shear resistance; $\gamma_t$ is the wet unit weight of soil; $\gamma_{sat}$ is the saturated unit weight of soil; $\gamma_w$ is the unit weight
of water; $h_1$ is the depth from the ground surface to the groundwater level; $h_2$ is the depth from the groundwater level to the
slip surface, which may correspond to the surface of the base layer; $H$ is the depth from the ground surface to the wetting
front; $\theta$ is the slope inclination angle; $c'$ is the cohesion of soil; and $\varphi'$ is the angle of shear resistance of soil.
**5 Slope stability analysis before and after the 2015 bushfire**
**5.1 Failure mechanisms of post-fire slope**
Based on the results of burning tests in Sect. 3, bushfires significantly affect soil properties, causing hazards such as hillslope
runoff, debris flows, and shallow landslides (Culler et al., 2023). A key effect is changes in soil water repellency, referring to
the inability of water to wet or infiltrate dry soil. This repellency may be strengthened or diminished depending on the
timescales post-fire (Varela et al., 2015). During the initial post-fire period, increased water repellency can occur a few
centimeters below the surface due to the formation of an impermeable layer from ash particles clogging soil micropores
(Mallik et al., 1984). However, soil water repellency changes over time. Strong winds can clear bushfire ash within days, and
rainfall can wash away the hydrophobic layer, increasing infiltration capacity (Pereira et al., 2013; Liu et al., 2021). Another
significant impact is the loss of root reinforcement. The losses lead to a decrease in shear strength by 55 %-82 % in the
months or years following a fire, and this decrease persists depending on fire severity, plant resistance, and regeneration rate
(Lei et al., 2022). It has been found that soil infiltration capacity can recover about a year after a fire, but the effects of root
loss continue to dominate, reducing tensile strength, hydrophobicity, and shear strength (Lanini et al., 2009).

Considering the post-fire timescales, failure mechanisms during heavy rainfall events can be categorized into two patterns.
The first pattern occurs shortly after a fire when soil water repellency is dominant. Vegetation losses and changes in soil
texture increase infiltration capacity, but ash from burned vegetation can block soil pores, forming a sealing layer. This layer
causes rainwater to accumulate, resulting in runoff and erosion. The second pattern, occurring after a longer period, is due to
decreased soil strength from reduced root systems. Reduced roots decrease transpiration and soil suction, lowering shear
strength (Ng and Menzies, 2014). Rising groundwater levels after rainfall elevate pore pressure, triggering slope failures or





large-scale landslides. This mechanism is related to root cohesion, with the depth and concentration of fire-damaged roots
influencing the sliding surface position. This study focuses on a post-fire timescale of approximately ten months, aligning
with the second pattern of failure mechanism, where root reduction predominates, leading to decreased soil shear strength.

**5.2 Study area and data used for analysis**

According to the Köppen-Geiger classification, the study area has a maritime temperate climate with warm summers and
cool winters, and 60 %-65 % of the annual rainfall occurring from May to October (Linforth, 1977). The vegetation in the
study area is classified as eucalypt open forest, a key resource for the timber industry (Specht, 1970). Approximately 48 % of
fires larger than 1000 ha from 2006 to 2016 occurred in eucalypt open forests. The geology of the study area is identified as
the Early Cretaceous Eumeralla Formation, Otway Group, dominated by sandstone lithology (Edwards et al., 1996). This
geological unit is known for its high landslide susceptibility in southwestern Victoria due to rapid weathering. The soil type
is classified as Kandosols, mainly found in areas underlain by Cretaceous sediments (Isbell, 2016). Kandosols, which lack
strong texture contrast between surface and subsoil horizons and often have weak or no structure, good drainage, and
permeability, are prone to sliding on steep slopes (WGCMA, 2008). The surrounding topography of the study area is steep,
featuring highly dissected bedrock knolls and ridges. Fig. 10 shows the distribution maps of elevation and slope for the study
area, respectively. Soil depth significantly influences the mechanism of shallow slope failures. In this study, soil depth
distribution ($y_i$) is calculated based on Eq. (19) proposed by Saulnier et al. (1997), which shows an inverse correlation
between soil thickness and slope angle.
$$y_i = y_{max}\left[1 - \frac{tan(x_i)-tan(x_{min})}{tan(x_{max})-tan(x_{min})}(1-\alpha)\right],\qquad\qquad(19)$$
Where $\alpha = y_{min}/y_{max}$; $y_{min}$ and $y_{max}$ are the minimum and maximum values of effective soil depth, respectively; $x_i$ is the
slope angle at element $i$; and $x_{max}$ and $x_{min}$ are the minimum and maximum values of slope angle, respectively.
In this study, the soil depth settings for calculations differ before and after the 2015 bushfire. Based on the soil depth map of
Australia (Rossel et al., 2014), the pre-fire minimum and maximum soil depths were set at 0.1 m and 2.0 m, respectively.
According to the second failure mechanism outlined in Sect. 5.1, the death and decay of tree roots after the fire reduce shear
strength, and the depth and concentration of fire-damaged roots influence the sliding surface position. Since the maximum
root concentration is at 0.3 m, significantly impacting soil shear strength (Baldwin and Stewart, 1987), the post-fire
minimum and maximum soil depths were adjusted to 0.1 m and 0.3 m, respectively. Fig. 11 shows the distribution maps of
soil depth for analysis before and after the bushfire in the study area. Soil parameters for analysis before and after the fire are
detailed in Table 2, based on the test results in Sect. 3 and adopted from the study by Rawls et al. (1983). Rainfall data were
sourced from the Lorne (Mount Cowley) station, published by BoM, located 12 km from the study area, covering the 72-
hour period from 12 to 14 September 2016 (Fig. 11). The 24-hour cumulative rainfall on 14 September was approximately

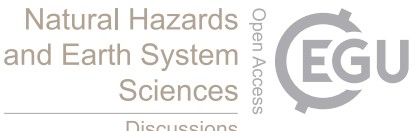

65 mm, consistent with the date and amount of rainfall observed during the post-bushfire landslides recorded by Colls and
Miner (2021). Due to the lack of detailed groundwater level information before the rainfall, the initial groundwater level is
assumed to be at the bottom of the soil depth.

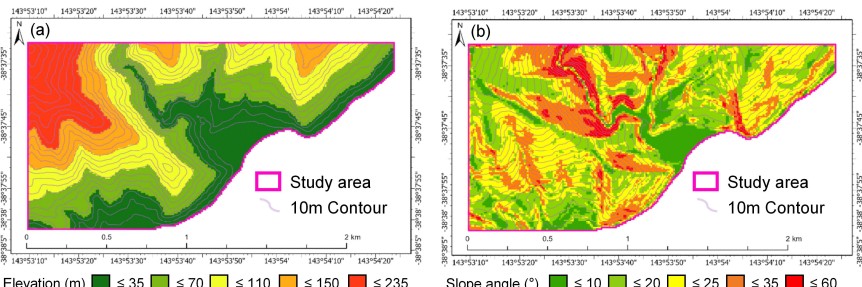


**Figure 10: (a) Elevation of the study area from 5 m DEM (5 m DEM provided by Geoscience Australia, 2015). (b) Slope angle of**
**the study area.**

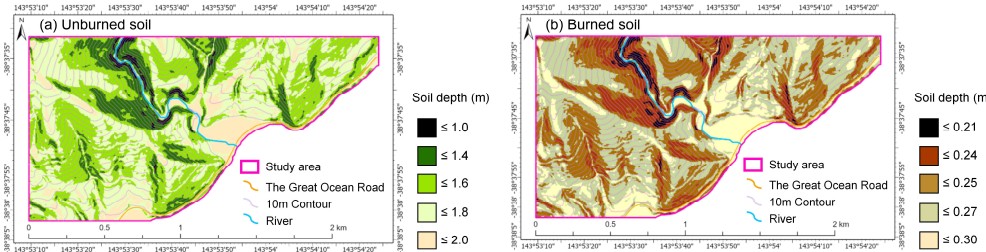


**Figure 11: Soil depth of the study area before (a) and after (b) the bushfire.**

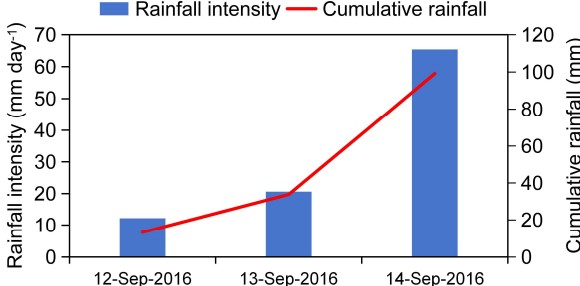


**Figure 12: Rainfall data from Lorne (Mount Cowley) rain station (provided by the Bureau of Meteorology).**





**Table 2: Material parameters for analysis.**

| Material parameters | Soil conditions | |
|---|---|---|
| | Unburned | Burned |
| Initial saturation degree, $S_{r0}$ (%) | 59.45 | 42.18 |
| Hydraulic conductivity, $K$ (m h$^{-1}$) | 7.78E-04 | 3.29E-02 |
| Saturated unit weight, $\gamma_{sat}$ (kN m$^{-3}$) | 19.0 | 17.6 |
| Wet unit weight, $\gamma_t$ (kN m$^{-3}$) | 17.1 | 14.2 |
| Cohesion, $c'$ (kN m$^{-2}$) | 6.41 | 0.45 |
| Internal friction angle, $\varphi'$ (deg.) | 39.33 | 40.24 |

**5.3 Results and discussion**
The slope stability analysis before the 2015 bushfire used the fine sand model, and the post-fire analysis employed the
coarse-medium sand model. As described in Sect. 4.3, unburned soil with low permeability exhibits a saturated layer from
the surface to the bedrock while burned soil with relatively high permeability shows the groundwater level rising from the
bedrock to the surface. Fig. 13a displays the maps of the wetting front depth from the ground surface before the fire from 12
to 14 September 2016. Fig. 13b shows the maps of the groundwater level depth after the fire, referring to the distance from
the ground surface to the groundwater level. Before the fire, at 00:00 on 12 September, no obvious distributions of wetting
front depth are observed in the study area. The distributions of wetting front depth begin to change significantly at 00:00 on
13 September and gradually become deeper with increasing rainfall. By the end of heavy rainfall at 00:00 on 15 September,
the wetting front depth reaches the bedrock surface in almost all slopes. After the fire, the groundwater level reaches the
ground surface in some slopes of the study area at 00:00 on 13 September and rises significantly with increasing cumulative
rainfall, reaching the ground surface in almost all slopes by 00:00 on 15 September. Due to the thinner maximum depth of
fire impact on surface soil (0.3 m post-fire vs. 2.0 m pre-fire), the post-fire groundwater level reaches the ground surface
earlier.

Figure 14 shows maps of surface water level depth from the ground surface before and after the fire, from 12 to 14
September 2016. Before the fire, surface water first appears near Separation Creek around 00:00 on 14 September, consistent
with the wetting front depth trend at the same time point (Fig. 13a). A larger area of surface water forms at 00:00 on 15
September. After the fire, some slopes show the groundwater level reaching the ground surface at 00:00 on 13 September,
making rainwater infiltration difficult and leading to surface water formation. Surface water tends to concentrate in valleys
and widen over time, appearing in almost the entire study area by 00:00 on 15 September. The areas of surface water
generally align with higher groundwater levels. Compared to pre-fire maps, post-fire surface water is distributed over a
larger extent and in more areas of high-level surface water.





The calculation formulas for the safety factor vary due to different infiltration behaviors (Eqs. 17 and 18). Figure 15 shows
the safety factor maps before and after the fire, from 12 to 14 September 2016. Before the fire, the values of safety factor ($F_s$)
decrease with increasing cumulative rainfall but remain above 1 due to the higher shear strength of the pre-fire surface soil,
indicating overall stability during the rainfall period. After the fire, from 00:00 on 13 September, the values of $F_s$ sharply
drop below 1 throughout the study area. By 00:00 on 15 September, the extent of $F_s$ below 1 significantly expands,
indicating potential slope failures, especially in steep slopes and river valleys. Figure 16 shows the magnified maps of $F_s$
before and after the fire at 00:00 on 15 September, respectively. According to Colls and Miner (2021), slope failures,
including the Paddy's Path landslide, occurred near the Great Ocean Road in the study area during heavy rainfall on 14
September. The results show that the values of $F_s$ of this road segment are above 1 before the fire but drop below 1 after the
fire, consistent with the locations of observed slope failures. Thus, the results are considered reasonably reliable and provide
a reference for predicting slope hazards in this area.

In some locations of this study area, especially near Separation Creek, the values of $F_s$ are less than 1, indicating instability,
however, no slope failures were observed. The lack of a detailed database on slope disasters complicates the identification of
slope failures in this study. Spittler and Wagner (1998) reported that regions susceptible to landslides often experience more
occurrences following bushfires. According to the landslide susceptibility map of Colac Otway Shire (AS Miner
Geotechnical, 2006), the segment of the Great Ocean Road where Paddy's Path landslide occurred is at the highest level of
susceptibility, and areas near Separation Creek have a moderate-high level of susceptibility. Thus this segment is considered
relatively high-susceptible to landslides after the fire, consistent with the results of safety factor maps. Differences in pre-fire
vegetation density also influence soil strength changes and recovery time. Goudie et al. (1992) noted that the highest bushfire
temperatures are often in densely vegetated areas. According to the burn severity map of the 2015 bushfire (Noske et al.,
2022), the area near the Great Ocean Road had high burn severity with majority crown burn, while the area near Separation
Creek had medium burn severity with majority crown scorch, understory burn, and some crown burn. For slope stability
calculations, the entire study area was simplified to high burn severity, which may have led to lower assessments of safety
factors near Separation Creek after the fire. However, Wondzell and King (2003) suggested that fires can accelerate
streambank erosion, potentially increasing slope failures. Thus, special attention should be given to potentially unstable
slopes near Separation Creek, and over-assessment of safety factors is reasonable for developing mitigation and prevention
strategies.

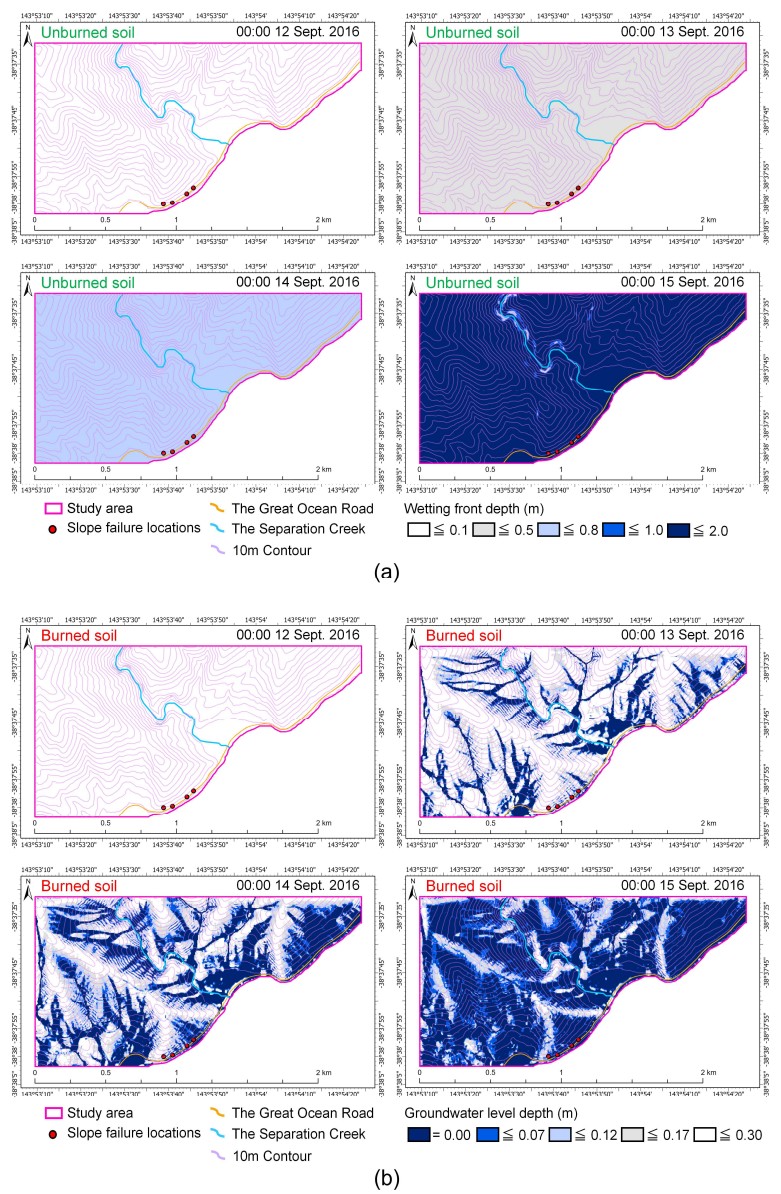


**Figure 13: (a) Maps of wetting front depth from ground surface pre-bushfire. (b) Maps of groundwater level depth from ground**
**surface post-bushfire.**


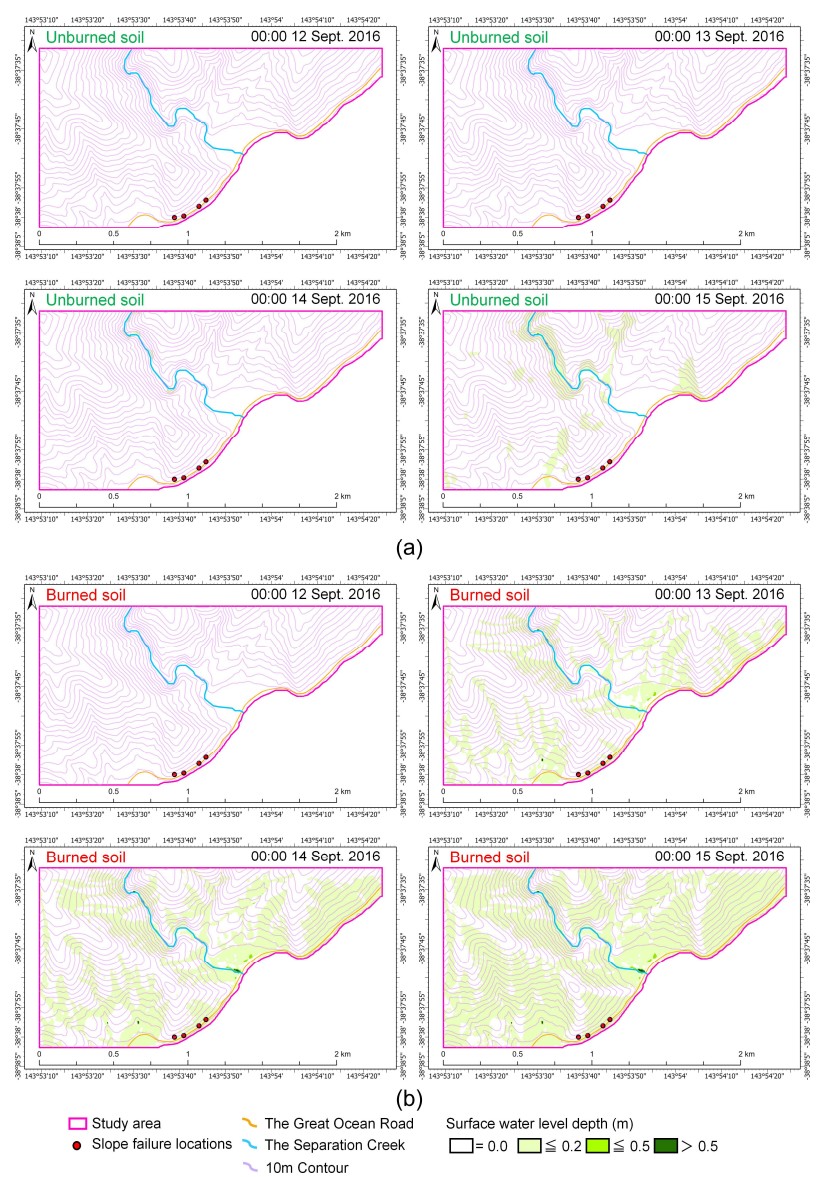


**Figure 14: Maps of surface water level depth from ground (a) pre-bushfire and (b) post-bushfire.**


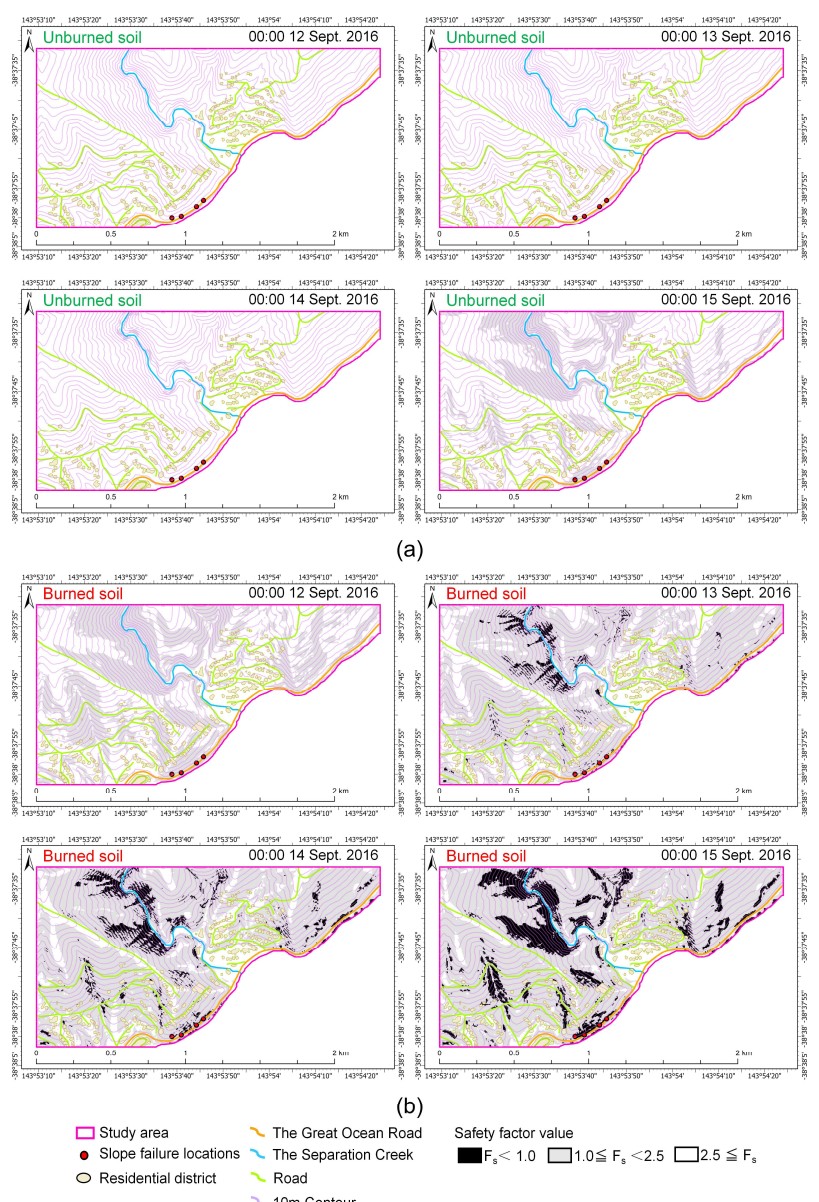


**Figure 15: Maps of safety factor (a) pre-bushfire and (b) post-bushfire.**


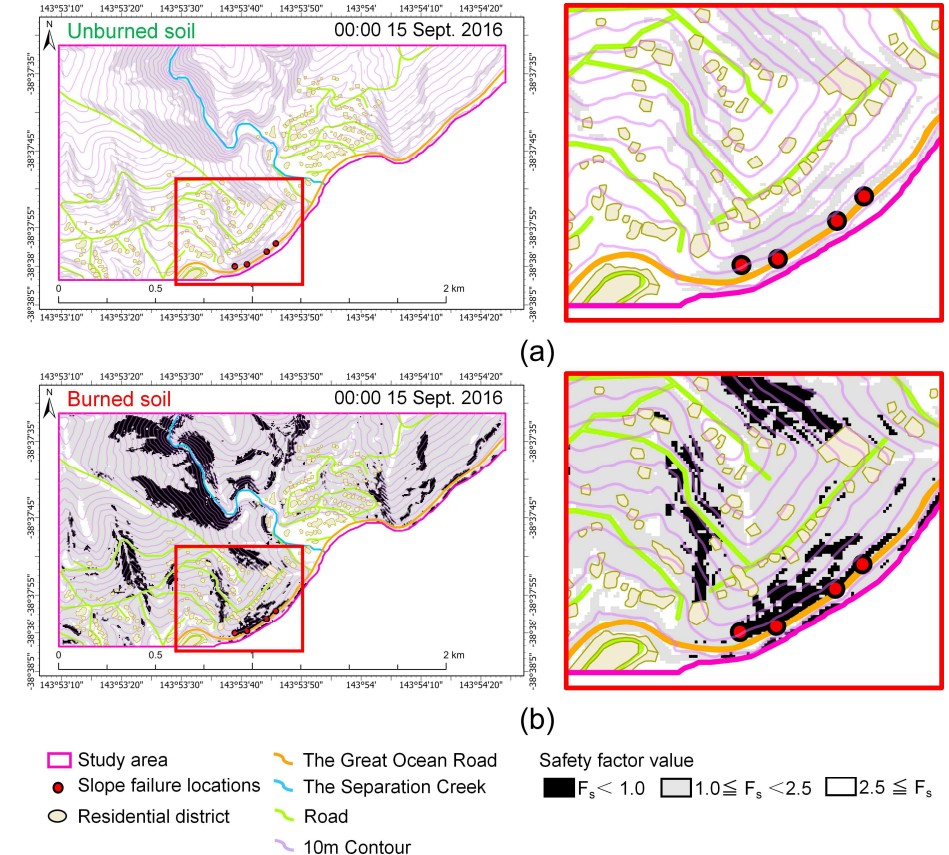

(a)

(b)

| Study area | The Great Ocean Road | Safety factor value |
| Slope failure locations | The Separation Creek | $F_s < 1.0$   $1.0 \leqq F_s < 2.5$   $2.5 \leqq F_s$ |
| Residential district | Road | |
| 10m Contour | | |


**Figure 16: Magnified maps of safety factor (a) pre-bushfire and (b) post-bushfire.**

## 6 Conclusions

This study focused on Wye River and Separation Creek in Australia, which were affected by the 2015 Wye River-Jamieson Track bushfire. Ten months after the bushfire, multiple slope failures were observed during heavy rains from 12 to 14 September 2016, including the Paddy's Path landslide, which disrupted the main road connecting the towns. This indicated a strong correlation between increased slope hazards and bushfire effects on soils. To determine this correlation, controlled laboratory burning tests were conducted to simulate the effects of bushfires on soil. The test results showed changes in the structural, hydrological, and mechanical properties of post-fire soil: coarser soil particles, increased permeability, and reduced soil shear strength, particularly cohesion. To compare changes in slope stability before and after the bushfire, using


a simplified hydrologic numerical model, slope stability assessments during rainfall were performed based on soil
parameters before and after the burning tests. The results confirmed significant differences in the time-dependent changes in
groundwater level depth, surface water depth, and safety factor during rainfall before and after the bushfire. Before the
bushfire, the values of the safety factor decreased with increasing cumulative rainfall but remained above 1, indicating stable
slopes in the study area. After the bushfire, due to weakened cohesion, it was observed that as the rainfall-induced
groundwater level rose, the values of safety factor fell below 1 in some slopes of the study area, indicating high susceptibility
to shallow slope failures. The results of the safety factor were below 1 at the locations where slope failures were observed,
confirming the accuracy of this study in capturing disaster occurrences. This study suggested that the dominant failure
mechanism of slopes is the reduction in shear strength due to the diminished root system after a bushfire, highlighting the
need for effective land management and disaster prevention measures. This information is valuable for future hazard
assessments and mapping landslide susceptibility after bushfires.
**Code and data availability**
Relevant code and data will be available to researchers upon request.
**Author contribution**
YL, AW, and SC conceptualized this study. YL conducted the experiments and carried out the analysis under the supervision
of AW and SC. YL prepared the manuscript draft, which was subsequently reviewed and edited by all co-authors.
**Competing interests**
The contact author has declared that none of the authors has any competing interests.
**Disclaimer**
Publisher's note: Copernicus Publications remains neutral with regard to jurisdictional claims made in the text, published
maps, institutional affiliations, or any other geographical representation in this paper. While Copernicus Publications makes
every effort to include appropriate place names, the final responsibility lies with the authors.



**Acknowledgements**
The authors are grateful to the Geoscience Australia for providing the DEM data and the Bureau of Meteorology for
providing rainfall data. The authors also appreciate the support of Ms. Saya Okabe, Mr. Kosuke Suto, and Ms. Mithushi
Wickramasinghe.

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
