# Peer review of "Bushfire effects on soil properties and post-fire slope stability: the case of the 2015 Wye River-Jamieson Track bushfire"

_Natural Hazards and Earth System Sciences, 2024_

## Author Comment (AC1)

We sincerely appreciate your thorough review and constructive comments on our manuscript. Your feedback
 has provided valuable guidance and will help improve the manuscript's structure and clarity. We have carefully
 considered each comment. Please find our detailed responses below, with author comments (AC) highlighted
 in blue.

**5 **RC2's comments:**

6 The manuscript presents a case study on the 2015 Wye River-Jamieson Trask fire. It aims to investigate 7 changes in soil properties after a fire and apply the measurements obtained to inform a hydrological model. 8 Soil samples were collected at unburned sites with similar characteristics to those at nearby burned sites. In 9 the lab, several soil tests were performed in the unburned samples and then, after burning them in a muffle 10 furnace, in the burned samples. The hydrologic model was used to evaluate slope stability during rainfall events 11 under unburned and burned conditions. The study is interesting and highlights the importance of informing 12 models with real measurements. Some of the laboratory test results are very interesting, and I suggest more 13 discussion of them should be developed. However, at this time the manuscript is a bit confusing, as it doesn't 14 follow a continuous line of thought. I suggest a major revisions and restructuring of the manuscript before 15 publication. 16 1. I would restructure the paper as follows:

17 1. introduction. 18 2. Materials and methods: 19 1. 2.1. Scope of the study - Study area (move also section 5.2. here) 20 2. 2.2. Soil sample collection 3. 2.3. Laboratory burning test conditions: Laboratory burning tests 21 22 4. 2.4. Numerical method: hydrological numerical method (name?) - model description and 23 parametrization for this specific area. 24 3. Results 4. Discussion 25 26 5. Conclusion 27 Line 129-191: should be in the result section. 28 Line 194-293: I suggest moving most of the equations to the supplementary materials (they take up a 29 lot of space and this article is not intended to present a new model) and reducing the model descriptions 30 to the essential information in the materials and methods sections. 31 AC: Thank you for reviewing our manuscript and offering valuable comments. We are pleased that you 32 recognize the significance of our work and appreciate your comments on expanding the discussion of the 33 laboratory test results and on restructuring the manuscript. In the revision, we will expand the discussion of 34 the experimental results. Furthermore, we will reorganize the manuscript as you suggested to enhance its 35 logical flow and readability by redefining the section divisions. We will reconstruct the content according to 36 the revised outline. Below is a brief description of the revised section titles and their contents: 1

- 38 2. Materials and methods
- 39 2.1. Scope of the study Study area (content from former Sections 2.1 and 5.2)
- 40 2.2. Soil sample collection (content from former Section 2.2)
- 41 2.3. Laboratory burning test conditions (content from former Section 3.1)
- 42 2.4. Hydrological numerical method (TAG\_FLOW): model description and parametrization for the study
- 43 area (concise description of the numerical model and introduction to parametrization for the study44 area; content from former Sections 4.5 and 5.2)
- 45 3. Results
- 46 3.1. Soil characteristics before and after the burning test (content from former Section 3.2, lines 129–191)
- 47 3.2. Slope stability analysis (content from former Section 5)
- 48 4. Discussion
- 49 5. Conclusion
- 50 References
- 51 Supplement (content moved from former Sections 4.1–4.4, lines 194–293)

**52 **Comments:**

Bushfire vs wildfire: wildfire is a general term that includes bushfires, and the literature you are citing
 is about wildfires. I would suggest changing the term since, as you mention at line 338, your study
 area is a large eucalypt open forest.

56 AC: Thank you for the constructive suggestion. We agree that wildfire is the more appropriate umbrella term

57 for this study. Accordingly, we will replace each instance of bushfire with wildfire throughout the manuscript

to align with the cited literature and more accurately describe that the study area is located at a large eucalypt

- 59 open forest.
- 60 2. In general, you could use more citation.
- 61 AC: Thank you for your valuable suggestion. We agree that additional citations will strengthen our manuscript.
- 62 We will review the relevant literature and add necessary references to support the arguments of this study.
- 63 3. Laboratory burning tests: you should be more clear about the number of samples you collected in the
  64 field and the amount of repetitions of each test you did. Did you measure properties only once per
  65 sample or did you do repetitions? Are the values in Table 1 the average?
- 66 AC: Thank you for pointing out this important issue. During field sampling on the same unburned slope, we
- 67 collected both undisturbed and disturbed soil samples. The undisturbed samples comprised: four cylinder
- 68 samples A (62 mm diameter, 20 mm height; see Figure R1a below), two cylinder samples B (82 mm diameter,
- 42 mm height; see **Figure R2a**), and one block sample  $(30 \text{ cm} \times 30 \text{ cm} \times 30 \text{ cm}; \text{see Figure R3})$ . Approximately
- 70 10 kg of disturbed soil was also collected (some of these samples are shown in Figure R4). Based on these

- samples, we conducted laboratory burning experiments and soil tests. The testing procedures and number of
- repetitions for the soil tests reported are explained in **Table R1** below. We will revise the manuscript to include
- respective respective

**74 Table R1: Testing procedures and number of repetitions for soil tests.**

|                                   | Soil conditions                                             |                                             |
|-----------------------------------|-------------------------------------------------------------|---------------------------------------------|
| 1 est results                     | Unburned                                                    | Burned                                      |
| Particle size distribution test   | Disturbed soil samples were tested                          | Disturbed soil samples after burning were   |
|                                   | twice; observed deviations were                             | tested using the same procedures as for     |
|                                   | within $\pm 3-5\%$ , and the mean of the                    | the unburned samples (see Figure R5b        |
|                                   | two measurements was used (see                              | below).                                     |
|                                   | Figure R5a below).                                          |                                             |
|                                   | Undisturbed soil samples consisted of                       |                                             |
| Moisture content,                 | four cylinder samples A; the mean of                        |                                             |
| w (%)                             | four measurements was used (see                             |                                             |
|                                   | Figure R1 below).                                           |                                             |
| Wet density,                      | As above (see Figure R1 below)                       |                                             |
| $\rho_t (\mathrm{g \ cm^{-3}})$   | As above (see Figure KI below)                              | 1                                           |
|                                   |                                                             | Undisturbed soil samples consisted of       |
| Dry density,                      | As above (see Figure R1b below)                             | two cylinder samples B; the mean of two     |
| $ \rho_d (\mathrm{g \ cm^{-3}}) $ |                                                             | measurements was used (see Figure R2b       |
|                                   |                                                             | below).                                     |
|                                   | Disturbed soil samples were tested                          | Disturbed soil samples after burning were   |
| Soil particle density,            | using three specimens; the mean of                          | tested using the same procedures as the     |
| $\rho_s (\mathrm{g \ cm^{-3}})$   | their results was used (see Figure R6a                      | unburned samples (see Figure R6b            |
|                                   | below).                                                     | below).                                     |
| Void ratio, e              | The value was calculated based on the                       | For simplicity, the moisture content of     |
|                                   | measured $\rho_s$ , $\rho_t$ and $w$ , using Eq. (1) | burned soil samples was assumed to be       |
|                                   | shown below (Lambe and Whitman,                             | zero; the same calculation method as for    |
|                                   | 1969).                                                      | undumed samples was appned.                 |
| Liquid limit,
$w_L(\%)$        | Testing was conducted on disturbed                          | Testing was conducted on disturbed soil     |
|                                   | soil; results from three specimens                          | after burning; the resulting particles were |
|                                   | were averaged (see Figure R7a                               | predominantly non-plastic sand, so the      |
|                                   | below).                                                     | measured liquid limit was zero (see         |
|                                   |                                                             | Figure R7c below).                          |
| Plastic limit, $w_p$ (%)          | As above (see Figure R7b below)                             | As above (see Figure R7c below)      |

| Plasticity index,
$I_p$ (%)                           | The value was calculated using Eq.
(2) based on the measured liquid limit
and plastic limit (Lambe and
Whitman, 1969).                        | As above (see Figure R7c below)                                                                                                                |
|----------------------------------------------------------|---------------------------------------------------------------------------------------------------------------------------------------------------------------|-------------------------------------------------------------------------------------------------------------------------------------------------------|
| Hydraulic conductivity,
K (m s -1 ) | The test sample was an undisturbed
block specimen; three replicate tests
were conducted, and the mean result
was used (see Figure R8a below). | A burning test was performed on the same block sample; testing followed the same procedures as for the unburned sample (see Figure R8b below). |
| Internal friction angle,                                 | Disturbed soil samples were tested in three replicates, and the mean value                                                                                    | For disturbed soil samples that were
burned, the same testing procedures as for                                                                    |
| $\varphi$ (deg.)                                         | was used.                                                                                                                                                     | the unburned samples were applied.                                                                                                                    |
| Cohesion,
c (kN m -2 )                     | As above                                                                                                                                                      | As above                                                                                                                                              |

(b) Undisturbed cylinder samples B after burning

Figure R2: Cylinder samples B used for measuring post-burning dry density.

---

## Author Comment (AC2)

We sincerely appreciate your thorough review and constructive comments on our manuscript. Your feedback
 has provided valuable guidance and will help improve the manuscript's structure and clarity. We have carefully
 considered each comment. Please find our detailed responses below, with author comments (AC) highlighted
 in blue.

**5 **RC1's comments:**

6 This manuscript presents a case study for a modeling framework that combines numerical models and 7 laboratory experiments to predict slope factor-of-safety values under pre- and post-fire conditions for a 8 landscape in Victoria, Australia, burned by the 2016 Wye River-Jamieson Track wildfire. This location 9 experienced shallow landslides in response to rainfall approximately 10 months after the fire was contained. 10 The authors predict factors of safety using a hydrological model that simulates subsurface and overland flow 11 given input rainfall, and they use controlled laboratory burn experiments on similar soils collected outside of 12 the burn area to parameterize certain model values. The results of their modeling experiments demonstrate a 13 widespread increase in slope instability as indicated by factors of safety less than 1 after the fire due to 14 increased soil saturation and diminished soil cohesion that overlaps with the location of observed landslides. 15 The focus of this study and the predictive nature of their methods are of scientific value and would be of 16 interest to the community.

However, this reviewer has multiple concerns that need to be addressed before publication can be recommended. These include model selection in the pre-fire versus post-fire cases, choice of model parameter values, details of the controlled laboratory burn experiments, a need for expanded literature review, and insufficient consideration of uncertainty. Each of these topics is described in greater detail below. If these can be satisfactorily addressed, as well as the line-by-line comments at the end of this comment, then this reviewer could recommend publication to the editor.

23 1. Model selection: two distinct subcategories of the model are chosen for the pre-fire and post-fire cases, 24 namely the "fine sand" model and the "medium-coarse sand" model. These selections are made on the 25 basis of measured grain size distributions and hydraulic conductivity in the controlled burn laboratory 26 experiment. However, it is unclear the impact that this choice alone has on the modeling results, as 27 different equations are used to calculate the factor of safety in these models (i.e., equation 17 for the post-fire case, and equation 18 in the pre-fire case). The post-fire factor of safety equation depends on 28 29 the depth to groundwater and the distance from the groundwater level to the slip surface. On the other 30 hand, the pre-fire factor of safety equation depends on the depth of the wetting front from the surface. 31 All else held equal, this reviewer wonders how this difference in physical process representation 32 impacts the modeling results.

AC: Thank you for your valuable comments. We have provided the following responses to the issues
 you raised and will revise the manuscript accordingly.

35 In this study, the fine sand model was applied under pre-fire conditions, whereas the medium-coarse 36 sand model was adopted post-fire. This selection was based on experimental results indicating that soil 37 particle size increased and permeability improved after the fire, thereby shifting the dominant 38 hydrological processes. Laboratory particle-size distribution tests indicated that the pre-fire soil was 39 composed of fine sand, making it appropriate to apply the fine sand model for slope stability analysis 40 (Ozaki et al., 2021; Wakai et al., 2019). This model focuses on the position and downward 41 advancement of the wetting front and is therefore suitable for low-permeability materials. During the 42 initial stage of rainfall, water is retained by capillary forces in the unsaturated zone, and the wetting 43 front gradually progresses downward. The groundwater level responds slowly, and full saturation does 44 not occur until the wetting front reaches the initial groundwater level. Consequently, the fine sand 45 model employs Eq. (1) to calculate the factor of safety  $(F_s)$ , emphasizing the influence of wetting front depth. After the fire, the soil was exposed to high temperatures, and laboratory tests showed an increase 46 47 in particle size and enhanced permeability. As a result, the soil was classified as coarse sand. In the 48 medium-coarse sand model, under higher-permeability conditions, rainfall supply exceeds capillary 49 retention. When under intense rainfall, water can penetrate the unsaturated zone in a short time, 50 significantly raising the groundwater level (Nguyen et al., 2022). The slope response is therefore 51 governed by groundwater-level rise, and the slip surface can rapidly reach saturation and failure 52 conditions. Accordingly, Eq. (2) is used to characterize the reduction in shear strength induced by the 53 rise in groundwater level. Both models are supported by published literature and documented 54 parameter tests, and each has clearly defined applicable material domains (Nguyen et al., 2022; Ozaki 55 et al., 2021; Wakai et al., 2019). Thus, they should not be interchanged.

56
$$F_s = \frac{\tau_f}{\tau} = \frac{c' + \gamma_{sat} \cdot H \cdot \cos^2 \theta \cdot tan \varphi}{\gamma_{sat} \cdot H \cdot \sin \theta \cdot \cos \theta},$$
(1)

$$F_{S} = \frac{\tau_{f}}{\tau} = \frac{c' + [\gamma_{t} \cdot h_{1} + (\gamma_{sat} - \gamma_{w})h_{2}]cos^{2}\theta \cdot tan\varphi'}{(\gamma_{t} \cdot h_{1} + \gamma_{sat} \cdot h_{2})sin\theta \cdot cos\theta}$$
(2)

58 where  $\tau$  is the shear stress due to the sliding direction component of gravity of soil;  $\tau_f$  is the shear 59 strength of soil, which is the maximum shear resistance;  $\gamma_t$  is the wet unit weight of soil;  $\gamma_{sat}$  is the 60 saturated unit weight of soil;  $\gamma_w$  is the unit weight of water;  $h_1$  is the depth from the ground surface 61 to the groundwater level;  $h_2$  is the depth from the groundwater level to the slip surface, which may 62 correspond to the surface of the base layer; *H* is the depth from the ground surface to the wetting front; 63  $\theta$  is the slope inclination angle; *c'* is the cohesion of soil; and  $\varphi'$  is the angle of shear resistance of 64 soil.

Following your suggestion, we conducted a sensitivity analysis: under the same rainfall and soil 65 66 parameters, we switched only the  $F_s$  calculation formula (fine sand model versus medium-coarse sand 67 model) to compare their effects on the  $F_s$  results. Under the pre-fire parameter conditions,  $F_s$  values 68 across the study area were calculated using both the fine sand model (the primary model in this study) and the medium-coarse sand model (Figure R1a and Figure R1b). The results from both models show 69 70 that  $F_s > 1$  across the entire area, with no unstable points. However, the absolute  $F_s$  values at the 71 same locations differed significantly between the two models, with an average difference of about 72 20% and a maximum difference exceeding 50%. This indicates that even when the slope stability 73 assessment results are consistent, the model structure has a substantial influence on the quantitative  $F_s$

74 values. Furthermore, as shown in Figure R2, the  $F_s$  results were calculated under post-fire parameter 75 conditions. Figure R2a and Figure R2b represent the simulation results from the fine sand model and the medium-coarse sand model (the primary model in this study), respectively. It can be observed that 76 77 the  $F_s$  values obtained using the fine sand model are all greater than 1, indicating that the entire area 78 is classified as stable. However, when compared with the zones where landslides were observed, the 79 fine sand model clearly underestimates the actual landslide hazard. In contrast, the medium-coarse 80 sand model identifies multiple points with  $F_s < 1$ , which correspond well with the locations of the 81 observed landslides.

82 Based on the above results, under pre-fire parameter conditions, the material properties remain far from the instability threshold. Therefore, all points were classified as stable ( $F_s > 1$ ), regardless of the 83 model structure chosen. However, after the fire, material strength decreased, and only the model that 84 correctly matched the underlying physical mechanisms (the medium-coarse sand model) can identify 85 the actual landslide locations, while the mismatched model (the fine sand model) underestimated the 86 87 hazard. Therefore, it is difficult to describe the soil changes examined in this study using a single unified formula. In this study, we propose to apply separate models for pre-fire and post-fire conditions 88 89 precisely to ensure that the physical mechanisms are properly matched.